# Protective Roles of Cytosolic and Plastidal Proteasomes on Abiotic Stress and Pathogen Invasion

**DOI:** 10.3390/plants9070832

**Published:** 2020-07-02

**Authors:** Md. Sarafat Ali, Kwang-Hyun Baek

**Affiliations:** 1Department of Biotechnology and Genetic Engineering, Bangabandhu Sheikh Mujibur Rahman Science & Technology University, Gopalgonj 8100, Bangladesh; sarafatbiotech@ynu.ac.kr; 2Department of Biotechnology, Yeungnam University, Gyeongsan, Gyeongbuk 38541, Korea

**Keywords:** abiotic stresses, Clp protease, defense, pathogen, protease, proteasome

## Abstract

Protein malfunction is typically caused by abiotic stressors. To ensure cell survival during conditions of stress, it is important for plant cells to maintain proteins in their respective functional conformation. Self-compartmentalizing proteases, such as ATP-dependent Clp proteases and proteasomes are designed to act in the crowded cellular environment, and they are responsible for degradation of misfolded or damaged proteins within the cell. During different types of stress conditions, the levels of misfolded or orphaned proteins that are degraded by the 26S proteasome in the cytosol and nucleus and by the Clp proteases in the mitochondria and chloroplasts increase. This allows cells to uphold feedback regulations to cellular-level signals and adjust to altered environmental conditions. In this review, we summarize recent findings on plant proteolytic complexes with respect to their protective functions against abiotic and biotic stressors.

## 1. Introduction

Plants are immobile organisms and may thus be exposed to dynamically changing environmental conditions including abiotic stress and pathogen invasion [1]. The primary environmental stressors such as high temperature, cold, drought, chemicals, and salinity exert detrimental effects on plants [2]. Plant cells can be damaged by these factors and subsequently experience osmotic and oxidative stresses which are referred to as secondary stresses [2]. Under such conditions, the most crucial function of a plant cell is to induce self-defense against the respective stressor. Such defense mechanisms may include qualitative and quantitative alterations of gene expression which may lead to modulations in certain pathways [3].

Global changes in gene expression in response to stress have been observed in numerous organisms [4]. Under stressful conditions, expression of some genes is upregulated, whereas that of others is downregulated. Some genes known as heat shock (HS) genes are rapidly upregulated during heat stress [5]. HS proteins function in two different ways: (1) as molecular chaperones that counteract adverse protein denaturation and aggregation and (2) as ubiquitination agents that target non-native or orphaned proteins for subsequent degradation [6]. Biotic and abiotic stressors typically cause protein dysfunction, and aberrant proteins represent considerable hazards to cell viability [7]. Various cellular processes are severely affected by aberrant or damaged proteins [8].

In all living cells, proteins are synthesized and may be degenerated within the original cell when the proteins are damaged, misfolded, mislocalized, or when they are no longer required [9]. Protein quality is maintained by degrading abnormally structured proteins stemming from mutations or metabolic damage [10]. It has been reported that protein quality control is crucial in non-dividing cells where accumulation of malfunctioned proteins is detrimental [11]. Furthermore, cells respond to damaged proteins by increasing their proteolytic activity in order to counteract toxic effects of damaged proteins [12]. Removal of undesired or damaged proteins caused by severe environmental stresses is a tightly regulated proteolytic process. Removal of non-functional proteins from cells is crucial for maintaining homeostasis and physiological metabolic activities. Therefore, proteolytic functions of proteases is particularly important during conditions of stress that induce damage or impairment of proteins [13]. 

Proteolytic enzymes are generally termed proteases, proteinases, or peptidases [14]. Intracellular proteolysis is carried out by two major proteolytic systems: (i) proteases and (ii) proteolytic complexes. Depending on amino acid determinants of catalytic sites or required metal co-factors, proteases are categorized in five classes, namely serine proteases (EC 3.4.21), cysteine proteases (EC 3.4.22), aspartic endopeptidases (EC 3.4.23), metalloproteases (EC 3.4.24), and threonine proteases (EC 3.4.25) [15]. Many of these proteases work independently, however, in proteolytic complexes such as eukaryotic and prokaryotic proteasomes, many proteases and their regulatory proteins form vast quaternary complexes with coordinated functions to effectively degrade undesired proteins [16]. The most common proteolytic complexes of plants are proteasomes in the cytoplasm and the nucleus, and Clp protease complexes in plastids and mitochondria [17]. 

Accumulating evidence suggests elevated gene expression and increased activities of proteases in response to abiotic and biotic stressors [18]. The protective roles of proteases during environmental stresses and pathogen invasion were reviewed elsewhere [19,20,21]. A number of proteases in plastids are well characterized and were found to be localized in different compartments such as the stroma, thylakoids, and the lumen [22]. ATP-dependent caseinolytic (Clp) protease complexes are multi-subunit complexes that are important for protein degradation in plastids, and plastid proteases appear to be constitutively expressed; however, their expression can be induced in response to certain environmental stressors [23]. Inductive mechanisms during stressful conditions include expression of serine protease due to biotic stress [24]. Plastids can experience severe damage due to environmental and biotic stressors including denaturation of proteins within them [25]; however, a comprehensive review on the effects of Clp protease on stress responses and resistance is lacking so far. Recovery or salvage of denatured plastidal proteins after stress is likely vital for plants to maintain tolerance of numerous environmental stressors including pathogens.

There are vast groups of proteases that are induced by abiotic and biotic stressors, therefore, in this review, we focus on recent findings of plant proteasome complexes with respect to both the prokaryotic and the eukaryotic type. Clp protease complexes in plastids originate from prokaryotes and exert protective roles against abiotic and biotic stressors. 26S proteasomes in the cytoplasm and nucleus also regulate stress responses, thereby increasing resistance. After comparing the structures of these two different types of proteasomes, proteasome functions in response to various abiotic and biotic stressors are explained. 

## 2. 26S Proteasomes in Plants 

Proteasomes are highly conserved protein complexes that occur in all eukaryote cells and in archaea [26]. Proteasomes degrade non-native or damaged proteins by proteolysis, thereby regulating the concentration of particular proteins. Such degradation processes result in short peptides of about seven to eight amino acids, and these short peptides are further degraded to amino acids which are used for synthesizing new proteins [27]. Maintaining a tightly coordinated and highly specific system for degradation of individual proteins is crucial for the survival of any organism. In eukaryotes, this is accomplished by tagging target proteins with ubiquitin for subsequent recognition and degradation by the 26S proteasome. The ubiquitin proteasome pathway is predominant in the cytoplasm and nucleus to protect the cell from toxic effects of misfolded proteins [28].

The 26S proteasome contains one 20S core particle and two 19S regulatory caps (Figure 1). The 20S particle is barrel-shaped and consists of four stacked heptagonal rings. The two inner rings are β-catalytic rings (β1-7), and the two outer rings are α rings (α1-7). The α rings serve as an interface for regulatory particle (RP) binding, and the α subunit N-termini form a gate that prevents unregulated access to the interior cavity [29].

19S RPs are composed of two sub-complexes. The sub-complex which is proximal to the 20S catalytic particle is termed ‘base’, and the distal sub-complex is termed ‘lid’. The base contains six ATPase subunits (Rpt1 to Rpt6) and four non-ATPase subunits (Rpn1, Rpn2, Rpn10, and Rpn13). The lid contains eight non-ATPase subunits (Rpn3, Rpn5 to Rpn9, Rpn11, and Rpn12**)** [9,30]. Rpn10 and Rpn13 serve as linkers between lid and base [9]. Orphaned or damaged proteins are recognized and unfolded by the lid and are then fed to the 20S proteolytic core by the base [31]. 26S proteasomes work in a precise manner. To prevent accidental capture of non-substrate proteins, the target proteins are tagged by covalent attachment of polyubiquitin chains (Figure 1). The consecutive action of ubiquitin-activating (E1), -conjugating (E2), and -ligating (E3) enzymes are essential for adequate ubiquitination [28,32].

Non-lysosomal protein degradation which is mediated by the 26S proteasome pathway in the nucleus and cytoplasm is fundamental for regulating diverse cellular processes [33]. The 26S proteasome plays a key role in various cellular processes by disintegrating short- and long-lived proteins. About 80–90% of the bulk of cellular proteins are disintegrated by the 26S proteasome. Even misfolded proteins in the endoplasmic reticulum (ER) go through a retrograde transport from the ER to the cytosol for subsequent degradation by the 26S proteasome [34]. Therefore, inhibition of the 26S proteasome causes severe and rapid loss of universal protein synthesis [35].

## 3. The Clp Protease System in Plastids

Plant cell plastids comprise an important proteolytic system where proteases play a crucial role in respect of precursor proteins and degradation and removal of undesired or damaged proteins. ATP-dependent proteases combine chaperones with peptidase activity, and chaperone activity is paramount for unfolding protein substrates and feeding them to a protein degradation chamber where peptidolysis occurs [36]. There are three major types of ATP-dependent proteases in plastids and mitochondria of eukaryotic plant cells which were inherited from their eubacterial ancestors, i.e., the ATP-dependent Zn-metalloprotease FtsH family [37,38], the ATP-independent Deg/HtrA family of serine endopeptidases [39,40,41,42], and the ATP-dependent serine-type Clp family [38,43]. FtsH family members are located at the thylakoid or at the inner envelope membranes and, particularly, at plastid thylakoids [44,45,46]. The Deg proteins are ATP-independent serine-type proteases which occur as both lumenal (Deg1, 5, and 8) and stromal compounds (Deg2 and 7) in plastids and play a crucial role in photosystem-II assembly [39,41,42,47]. The Clp family members which are important for protein degradation are confined to the stroma with some occurring at the chloroplast membranes [38,43,45]. 

The Clp protease system was first identified in *Escherichia coli*, and it consists of two components, namely the proteolytic subunit (ClpP) and the ATPase subunit. The ATPase subunit belongs to the AAA+ protein family which, consisting of eight subfamilies (ClpA, B, C, D, M, N, X, and Y/HslU), has been identified across various species [48,49,50,51]. Depending on the number of nucleotide binding domains (NBDs), Clp proteins are categorized in two classes [52]: class I proteins (e.g., ClpA, ClpB, ClpC, and ClpD) which are relatively large (68–110 kDa) and have two NBDs and class II proteins (e.g., ClpM, ClpN, ClpX, and ClpY/HslU) which are comparably small (40–50 kDa) and have one NBD. 

ClpA has been detected in Gram-negative bacteria, whereas ClpB has been detected in prokaryotes (known as ClpB), yeast (Hsp104), and plants. ClpB in plants occurs in the cytosol (known as ClpB-C), in mitochondria (ClpB-M), and in chloroplasts (ClpB-P) [53]. ClpC occurs in Gram-positive bacteria, cyanobacteria, and in chloroplasts of algae and higher plants, whereas ClpD (also referred to as Erd1) is restricted to the chloroplasts of higher plants. ClpM has been found in *Mus musculus* and *Plasmodium falciparum*, ClpN occurs in *Pseudomonas aeruginosa*, ClpX was found in bacteria, humans, and higher plants, and ClpY occurs in bacteria [54,55].

The Clp protease system in higher plants is diverse. In *Arabidopsis thaliana*, Clp proteases comprises more than 15 proteins with three HSP100 AAA+ chaperones (ClpC1, ClpC2, and ClpD), five serine-type Clp proteolytic subunits (ClpP1, ClpP3, ClpP4, ClpP5, and ClpP6), two adapter proteins (ClpS1 and ClpF) to bind ClpC, four non-proteolytic administrative or regulatory subunits (ClpR1, ClpR2, ClpR3, and ClpR4), and two proteins (ClpT1 and ClpT2) that function as stabilizers of the ClpRP core [17,38,43,56,57,58,59] (Figure 2). All genes encoding Clp protease subunits are part of the nuclear genome, apart from the *ClpP1* gene which is part of the plastid genome. 

The various proteins of the Clp protease machinery constitute two oligomeric components, namely (i) a tetradecameric barrel-shaped protease core with its catalytic sites within the complex and (ii) an ATP-dependent hexameric ring of chaperones. The chaperone ring recognizes non-native or damaged proteins with or without the help of adaptors and then un-bends these proteins and translocates them into the proteolytic chamber for degradation [43,60]. During and after stress conditions or during normal growth, the fate of any given denatured or misfolded protein in chloroplasts and mitochondria is determined by the Clp chaperone system. Therefore, degradation of orphaned or damaged proteins, stress responses, and gene regulation by proteolysis of transcription factors are the functional role of Clp complexes. 26S proteasome in the cytosol and nucleus and Clp proteases in the plastids and mitochondria show considerable similarities regarding structure and function, and we therefore compared them as shown in Table 1.

## 4. Roles of Plant Proteasomes in Response to Stressors and Pathogens 

Most of the cytosolic and nuclear proteins are processed by the 26S proteasome system. Ubiquitin binds to a protein at its lysine residue and thereby tags it for degradation by the 26S proteasome system. Misfolded or aberrant proteins and regulators of numerous processes are degraded by the ubiquitin-proteasome system. Ubiquitin/26S proteasome-mediated proteolysis is crucial in numerous cellular responses such as those associated with biotic and abiotic stress tolerance [61], pathogen defense [62], hormone signaling [63], morphogenesis [64], and chromatin modification [65]. 

Changes in proteasome abundance which are affected by development and environment are important for plant development and survival under adverse conditions [66]. As cell proliferation in plants depends on optimal 26S proteasome activity, stressors that directly affect 26S proteasome activity were suggested to indirectly reduce cell proliferation. Abiotic stress inhibits 26S proteasome activity either by decelerating the turnover rate of other 26S proteasome targets, by increasing the substrate load, or by directly affecting 26S proteasome functions. The substrates are proteins produced due to HS and other stresses that cause protein misfolding. Oxidative stress directly leads to 26S proteasome inhibition [67,68,69].

The ubiquitin-proteasome pathway in plants controls a range of cellular signaling processes, such as those elicited by hormones, sucrose, and light, as well as development and responses to pathogen invasion [2,70]. E3 ubiquitin ligases mediate the final transfer of ubiquitin to target proteins, which is a vital part of the degradation process. Furthermore, numerous studies showed the involvement of E3 ubiquitin ligases in plant defense systems [71]. In the *Arabidopsis* genome, approximately 1300 genes encode a E3 ubiquitin ligase motif [72]. Microarray screening data of in silico expression analyses on all annotated E3 ubiquitin ligase components revealed that biotic stress caused upregulation of up to 548 E3 ubiquitin ligase components and downregulation of 382 of such components [28]. E3 ubiquitin ligases and associated protein breakdown are vital for signal transduction pathways associated with disease resistance [28,73,74], and they are involved in plant defenses through controlled proteolysis during mechanisms associated with gene-for-gene disease resistance, early-defense response reactions, and late-induced defense responses [19]. 

Plant cells may evolve more elaborate molecular mechanisms under high-intensity stress and may alter 26S proteasome activity in response to variations in environmental conditions. This type of mechanism depends on the ubiquitin proteasome system [75,76,77]. The hot pepper (*Capsicum annuum* L.) U-box protein 1 (CaPUB1) and its *Arabidopsis thaliana* homologues AtPUB22 and AtPUB23 are ubiquitin ligases. During stress caused by abiotic factors such as desiccation, cold, or mechanical wounding, expression of the respective genes CaPUB1, AtPUB22, and AtPUB23 is rapidly induced [75,78]. In *C. annuum* and *A. thaliana,* PUBs ubiquitinate specific subunits of the RP lid sub-complex and interfere with functions of the 26S proteasome. CaPUB1 ubiquitinates Rpn6 and destabilizes the RP subunit [78]. Rpn12a is ubiquitinated by AtPUB22 and AtPUB23, which leads to relocation of a portion of Rpn12a to a cytosolic complex reminiscent of the proteasome-related 500-kDa complex (PR500). The PR500 complex contains the subunits of the RP lid and occurs as a stable separate particle in plant cells under physiological conditions (i.e., in unstressed plants). During heat stress and treatments with the amino acid analog canavanine, PR500 is depleted [79]. PR500 is used by plants to accelerate 26S proteasome biogenesis, which is required for ameliorating adverse effects of protein misfolding due to stress, particularly desiccation stress [79]. 

During drought stress, 26S proteasome levels are reduced due to the effect of AtPUB22/23 action by redirecting a portion of RP subunits to the PR500 particle. A reduction in 26S proteasome activity is detrimental for plant survival because desiccation tolerance depends on the ubiquitin-proteasome system [80]. Overexpression of AtPUB22/23 indicates hypersensitivity to drought stress, whereas loss of function suggests drought tolerance [75,78]. Both ligases are induced during stress, which indicates that they are required by plant cells to ameliorate adverse effects of stress. Loss of function of the 26S proteasome results in decreased root growth [81]. AtPUB22/23 overexpression elicits increased root elongation, which may indicate other functions in addition to AtPUB22/23 effects on 26S proteasome biogenesis. 

Proteasomes are involved in plant defenses against pathogen invasion [81,82,83]. The proteasome activity was required in cucumber hypocotyls (*Cucumis sativus*) for elicitation of defense responses [84]. Tobacco (*Nicotiana tabacum*) plants treated with cryptogein (a proteinaceous elicitor secreted by a fungal pathogen) upregulated expression of genes encoding b1-tcI 7, α3, and α6. [85,86]. During the induction of systemic acquired resistance and production of reactive oxygen species (ROS), expression of 20S subunits is increased [87]. In tobacco plants treated with cryptogein, production of ROS increases, which is mediated by the NADPH oxidase and elicits accumulation of β1 din 20S subunits [88]. Studies on loss or gain of function of β1 din 20S subunits showed that during elicitation of plant defense reactions, a proteasome consisting of a β1 din 20S subunit acts as a negative regulator of NADPH oxidase and contributes to the regulation of ROS generated during pathogen invasion [88].

Disruption of 26S proteasome function alters the ability of plants to efficiently and effectively respond to and tolerate various environmental stressors. Mutations of RP components affect 26S proteasome functioning, resulting in reduced complex accumulation, reduced rates of ubiquitin-dependent proteolysis, and modifications in responses to abiotic stressors [70,81,89,90]. *Arabidopsis rpn1a-4*, *rpn1a-5*, and *rpn10-1* mutant plants show limited tolerance to salt stress [89,91], and *rpn10-1* mutant plants are also hypersensitive to DNA-damaging agents and UV radiation [89]; *rpn1a-4, rpn1a-5, rpn10-1, rpn12a-1,* and *rpt2a-2* mutants show reduced HS tolerance [81,91], and *RPT2a* and *RPT5a* mutant plants are less tolerant to zinc-deficiency [92]. Results of these studies on mutant plants emphasize that the 26S proteasome plays a pivotal role for plant responses to adverse growth conditions.

Ubiquitin and ubiquitin enzymes are also important for plant responses to abiotic stressors. Most ubiquitin genes are expressed during stress [93,94,95]. Overexpression of monoubiquitin or polyubiquitin genes increase tolerance of plants to multiple abiotic stressors including salinity, cold, and drought [80,96]. Expression of E2 enzymes is differentially regulated in response to abiotic stressors. Among 39 E2-encoding genes (*OsUBCs*) in rice (*Oryza sativa*), expression of 14 genes was either upregulated or downregulated when plants were subjected to drought and/or salt stress [97]. Overexpression of mung bean (*Vigna radiata*) *VrUBC1*, soybean (*Glycine max*) *GmUBC2*, and peanut (*Arachis hypogaea*) *AhUBC2* in *A. thaliana* increased tolerance to drought stress [98,99,100]. Expression of the *NtUBC1* gene in tobacco increased in response to cadmium stress [101]. These observations suggest that 26S proteasomes in the cytoplasm and nucleus are critical for modulating the levels of regulatory proteins and for removing orphaned or non-native proteins in response to biotic or abiotic stressors.

## 5. Roles of Clp Protease Complexes during Stress and Pathogen Defense

In the cytosol and nucleus of plant cells, non-native or damaged proteins are degraded by 26S proteasomes, whereas in chloroplasts and mitochondria, this function is performed by prokaryote-type proteases due to the absence of proteasomes. The major protease in chloroplasts is the ATP-dependent stromal Clp proteolytic complex [17,43,59]. Expression of Clp proteases has been reported during drought, salinity, osmotic shock, pathogen invasion, oxidative stress, heat, and cold [102,103].

### 5.1. Roles of Clp Protease Complexes in Bacteria during Stress Conditions

Aberrant and denatured proteins are accumulated under stress conditions. Cells respond to this by increasing the synthesis of a set of highly conserved chaperones and proteases which either refold or degrade damaged proteins. In bacteria, ClpP-ClpA proteases are involved in the degradation of misfolded proteins [104,105]. ClpBs resolubilize protein aggregates during HS and other stresses [106]. ClpB is substantially induced in the unicellular *Synechococcus* sp. strain PCC 7942 during moderate cold stress [107], and the *ClpC* gene of *Bacillus subtilis* is induced in response to various stressors including cadmium stress [108]. In *Staphylococcus aureus*, expression of ClpB, ClpL, and ClpCP increases during heat stress, while ClpXP increases during osmotic stress, oxidative stress, and cold stress [109]. The presence of ClpC, ClpP, or ClpX in the cell is indispensable for stress tolerance, and protein levels of ClpC, ClpP, and ClpX increase during heat stress in *B. subtilis* [110]. Stress induction of ClpP in *E. coli* was first shown during heat shock [111], when Clp protease degraded aggregated proteins in vivo [112]. During starvation, ClpP proteases increase their activity which is directed against certain carbon starvation proteins [113]. ClpP proteases play significant roles in stress tolerance by degrading misfolded proteins in *Porphyromonas gingivalis* [114] and *Actinobacillus pleuropneumoniae* [115]. Therefore, Clp proteases that degrade misfolded and damaged proteins are likely important for bacteria to survive during adverse environmental conditions.

### 5.2. Roles of Clp Protease Complexes in Land Plants during Stress Conditions

Clp proteases are constitutively expressed in various plant tissues, and they are most abundant in chloroplasts of green leaves. Molecular chaperones cooperate in vitro as part of a functional network under stress conditions during which chaperones prevent accumulation of misfolded proteins and actively assist in their refolding [116]. ClpB, ClpC, and ClpD subunit proteins work as molecular chaperones that help protect cellular proteins from stress by delivering client proteins to Clp proteases. The *A. thaliana* organelle *ClpB* genes in chloroplasts and mitochondria show constitutive expression levels, which increase during high temperatures [117]. Similarly, organelle *ClpB* genes in rice show low constitutive expression levels which are upregulated during or after heat stress [118]. Chloroplast *ClpB* genes are also expressed constitutively in lima beans (*Phaseolus lunatus*), and their expression levels are significantly upregulated at high temperatures [119]. In the *clpr2-1* mutant, chloroplast ClpB3 was greatly upregulated in both young and mature leaves [120].

In chloroplasts, constitutive ClpD levels are comparatively low. The ClpD protein is encoded by the gene *ERD1* (early responsive to dehydration 1) [121], and its expression increases due to high salinity, dehydration, dark-induced etiolation, cold, and senescence [55,122,123,124,125]. In *A. thaliana*, long periods of cold increased ClpD protein content in leaves [122,126].

There are some contradictory reports regarding the levels of mRNA and protein of ClpC under different stress conditions or at different developmental stages [126,127]. Following short-term stress, mRNA and proteins levels of ClpC did not change [122], whereas after intensive-light treatments for 2.5 h, the transcript levels increased [128]. In vivo trapping studies for the discovery of the substrates for Clp proteases also revealed that most ClpC components are involved in the stress responses [129].

In rice seedlings, the ATP-binding subunit of ATP-dependent Clp protease responds to cold stress [130]. Proteomics revealed that ClpC levels significantly increased during cadmium stress in tobacco [131]. Protein levels of ClpC were also high in *Amaranthus hybridus* L. roots under cadmium stress [132], suggesting that ClpC may be important for ameliorating toxic effects in plants.

Co-suppression of *ClpC1*/*C2* in *Nicotiana benthamiana* produced a phenotype with severe chlorosis, aberrant development, and growth retardation [133]. ClpC1 chaperones unfold proteins for Clp proteases, and their expression is substantially induced during senescence, suggesting altered specificity of this complex [134]. Inactivation of the *ClpC1* gene in *A. thaliana* reduced plant growth and hampered chloroplast development [135,136,137]. When mutants with different variations of Clp protease (*clpr1*, *clps*, *clpc1*, *clpc1*, *clpd*, *clpt1*, and *clpt2*) were treated with methyl viologen, *clpc1* and *clpc2* mutants were more resistant to methyl viologen, compared to other mutants [138]. *clpc1* and *clpc2* mutants show 90% similarity in DNA and amino acid sequences. Due to this sequence similarity, they likely compensate for adverse mutation effects reciprocally to degrade toxic aggregates following UV treatments. Therefore, Clp protease complexes are vital factors of plant survival during various conditions of environmental stress.

In *Chlamydomonas reinhardtii,* ClpP1 is associated with the deterioration of the thylakoid-bound subunits of cytochrome *b6f* and photosystem II complex [139,140,141]. The steady-state growth of *Cyanobacterium* is considerably affected by ClpP1 as it helps cyanobacteria to acclimatize to various environmental conditions. When these cyanobacteria were exposed to extremely intensive light, photoinhibition, or moderate but non-inhibitory lighting, the content of ClpP1 was significantly increased. Inactivation of ClpP1 in cyanobacteria of the genus *Synechococcus* produced pleiotropic changes during steady-state growth, but slower growth was observed at higher light intensities [142], indicating that protein turnover mediated by ClpP is crucial for cell division. The ClpP1 protein of *Synechococcus* is analogous to the chloroplast form rather than to bacterial ClpP. ClpP1 expression is strongly induced by UV-B or low-temperature treatments, and loss of ClpP1 substantially affects stress acclimation capacity in *Synechococcus* [143].

ClpP1 was found to be a prerequisite for shoot development in *Nicotiana tabacum* [144,145]. Loss of function of the *ClpP1* gene regarding the proteolytic subunit of Clp protease increased tolerance of rice seedlings to both ozone [146] and SO_2_ [147] treatments**.** Transcriptomics and proteomics revealed that ClpP5 was significantly increased in *Nicotiana tabacum* under salt stress [148], indicating that Clp proteases may be involved in the defense of plants against stressors.

The abundance of ATP-dependent Clp protease proteolytic subunits in leaves of maize (*Zea mays*) was increased in response to cold stress [149]. Expression of the proteolytic subunit was also upregulated in wheat (*Triticum aestivum*) stems during drought-induced senescence [150,151]. In *Rhazya stricta,* all genes of the proteolytic subunit (*ClpP*) were upregulated after 12 h of salt stress [152]. Increased mRNA and protein content of several ClpP isomers also occurred in *A. thaliana* during long-term cold and high-intensity lighting acclimation [122], and proteomics revealed that ClpP levels were significantly increased during cadmium stress in tobacco [131]. 

ROS are generated under both physiological and stress conditions in plasma membranes, chloroplasts, mitochondria, ERs, peroxisomes, and in the cell wall of plant cells. The major sources of ROS production during light conditions are chloroplasts and peroxisomes, whereas during dark conditions, mitochondria are the predominant producers of ROS in plants [153]. The Clp protease system protects the plants’ chloroplasts from ROS generated in the presence of light and from ROS generated due to environmental stimuli such as excess light, heat, water shortage, or nutrient starvation. Pulido et al. [138] reported that Clp protease systems contribute to plant survival under methyl viologen-triggered oxidative stress. 

Clp proteases are upregulated in senescing leaves and participate in the degradation of plastidial photosystem II [154], and they are involved in the degradation of damaged or surplus proteins in plastids [43]. Genes encoding Clp proteases were found to be upregulated during drought stress [155]. Taken together, Clp proteases are crucial for counteracting biotic and abiotic stress by degrading orphaned or non-native proteins. 

## 6. Concluding Remarks

Aggregation of proteins that are damaged owing to stress is frequently a cause of cell death. Clp proteases and proteasomes are important for protecting plant cells under adverse conditions. During normal growth or during and after stress conditions, the fate of a given misfolded, denatured, or non-native protein is determined by the proteasome and/or Clp protease machinery as they help the cell recover from various stresses either by repairing damaged proteins (protein refolding) or by protein degradation. In this way, proteasomes and the Clp protease machinery can restore protein homeostasis and promote cell survival in plants. Comprehensive identification of the ubiquitin proteasome system and of Clp proteases and their substrates regarding their diverse roles in cellular metabolism may be challenging, however, further structural and functional research is needed to fully explore these aspects and their role in diverse cellular pathways such as stress responses and hormone systems. This would provide important insights regarding plant resource utilization and adaption.

## Figures and Tables

**Figure 1 plants-09-00832-f001:**
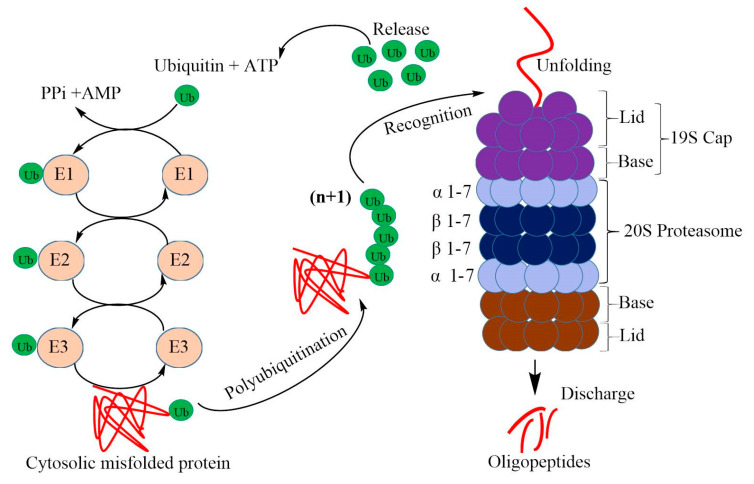
Schematic representation of ATP-powered proteolysis in the cytosol of plant cells. Proteins are tagged with multi-ubiquitin chains by the action of a series of ubiquitin ligases (E1, E2, and E3) and are targeted for degradation by the 26S proteasome. The 26S proteasome consists of the 20S proteasome sandwiched between two 19S regulatory particles. Upon binding of the protein substrate to the 26S proteasome, ubiquitin chains are recycled, and the protein is unfolded and degraded to oligopeptide fragments. E1: ubiquitin-ligase enzyme; E2: ubiquitin-conjugating enzyme; E3: ubiquitin-ligating enzyme.

**Figure 2 plants-09-00832-f002:**
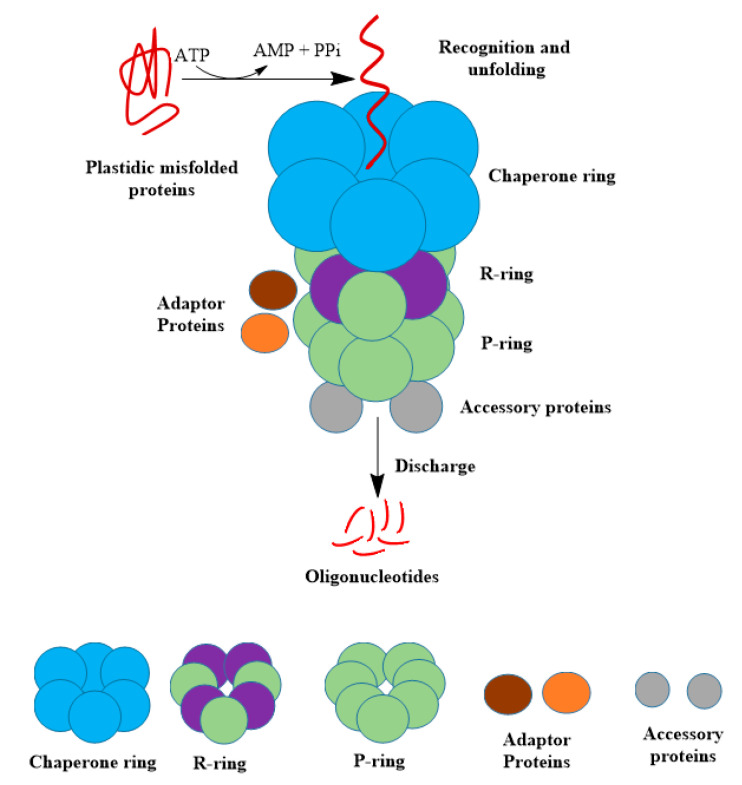
Schematic representation of ATP-powered proteolysis in plastids of plant cells. Misfolded proteins are recognized, unfolded, and translocated to the proteolytic chamber of the Clp protease complex by ClpC chaperones. The translocated proteins are degraded to oligopeptides that are exported through openings of the proteolytic chamber.

**Table 1 plants-09-00832-t001:** Comparison between 26S proteasomes and Clp protease.

Issues/Characteristics	26S Proteasomes	Clp Protease
Location	Cytoplasm and nucleus	Plastid and mitochondria
2. Components	19S regulatory particle and 20S core particle	Chaperone subunit and proteolytic subunit
3. Regulatory particle or Chaperone	19S regulatory particle, hexameric ring	Chaperone subunit, hexameric ring
4. Proteolytic subunit	20S core particle, heptagonal barrel-shaped ring	Proteolytic subunit, tetradecameric barrel-shaped ring
5. Components of Regulatory particle (RP) or Chaperone	Regulatory particle consists of base and lid. Base has six ATPase subunits (Rpt1-Rpt6) and two non-ATPase subunits (Rpn1 and Rpn2). Lid contains eight non-ATPase subunits (RPN3, 5–9, 11, and 12)	ClpC1, ClpC2, and ClpD constitute the chaperone subunit (in *Arabidopsis thaliana*)
6. Components of Proteolytic subunit	Four ring. Inner two rings are β catalytic rings (β1-7) and outer two rings are α rings (α1-7)	ClpP1, ClpP3, ClpP4, ClpP5, and ClpP6 constitute the proteolytic subunit (in *Arabidopsis thaliana*)
7. Linker or adapters	Rpn10 and Rpn13 serve as linker between lid and base	ClpS1 and ClpF serve as adapter for ClpC (in *Arabidopsis thaliana*)

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
