# Peer review of "Protective Roles of Cytosolic and Plastidal Proteasomes on Abiotic Stress and Pathogen Invasion"

_plants, 2020, doi:10.3390/plants9070832_

Round 1
Reviewer 1 Report
The review “Protective Roles of Cytosolic and Plastidial Proteasomes on Abiotic Stress and Pathogen Invasion” by Ali and Baek is a timely and informative contribution to the field of proteolysis.
The manuscript is well written and the style is concise.
Minor suggestions for improvement:
Fig. 1 is inferior. In particular, not even the transition from the seven-ring alpha and beta subunits to the six-ring ATPases is depicted in a sensible way. Fig. 2 uses different size balls, that is much better. Standard for depicting the proteasome is for instance the web page of the lab of Prof. Keiji Tanaka (Tokyo Metropolitan Institute of Medical Science).
Line 68 severe
Line 97 ff the description lists for the base two non-ATPase subunits (Rpn1,2), but later Rpn10 and 13 are called base subunits that link base to lid – so there would be two more non-ATPase base subunits to be mentioned.
Line 120 ubiquitin ligase not ubiquitin-ligating enzyme
Table 1 item 6 second column: four …rings
Line 210 ubiquitinate
Line 211 interfere
Reviewer 2 Report
The aim of the review “Protective Roles of Cytosolic and Plastidal Proteasomes on Abiotic Stress and Pathogen Invasion” is, according to the abstract, to “summarize recent findings on plant proteolytic complexes with respect to their protective functions against abiotic and biotic stressors” (Line 18).
In the introduction authors describe proteases and proteolytic complexes; the next two paragraphs 2-3, including the Figures 1-2 and Table 1, are dedicated to the detailed description of the structure and activity of 26S Proteasomes and Clp Protease System in Plastids.
The role of these complexes in protection against biotic and abiotic stress is discussed in the following paragraphs 5-6, but the lack of a proper organization of these paragraphs make it difficult to obtain a picture of the underlying mechanisms. In these paragraphs some general concepts are repeated in different parts (For example: Line 180-183 and Line 192-194) and specific results from different works are listed, but it is difficult to grasp a message. The subparagraph 5.1. “Roles of Clp Protease Complexes in Bacteria during Stress Conditions” generates further confusion in a review that is focused on plants.
As it stands, I think the manuscript needs major improvement to be of broad interest and hence suitable for publication. The general description of proteasomes and Clp protease systems should be summarized, while the discussion of their protective role on plant biotic and abiotic stress should be better developed and organized, with schemes and figures that could help the reader to understand the underlying mechanisms.
Reviewer 3 Report
The proposed review named "Protein malfunction is typically caused by abiotic stressors” regards recent findings on the roles of proteases for degradation of misfolded or damaged proteins within the cell with respect to their protective functions against abiotic and biotic stressors. I find the review written in good English. It gives us the opportunity to have an overall view on the involvement of the proteolytic system in the response of plants to stresses, therefore a valid basis for future studies.
I have only a minor comments
Page 2 ane 64 add the mean of the acronym Clp
Page 2 lane 72 remove invasion and write pathogens
Page 7 lane 228 Better explain the concept related to the following sentence "Competence development can be substantially suppressed by distinct inhibitors of proteasome activity in hypocotyls of cucumber (Cucumis sativus) [84]"
Round 2
Reviewer 2 Report
The article is centered on the structure and mechanism of action of 26S Proteasomes and Clp Protease Systems (illustrated also by Figures 1-2 and Table 1). The role of these complexes in the protection against biotic and abiotic stress is also discussed, although I found that this part (in particular for Clp complexes), mainly based on correlations with gene expression (increase of expression of different Clp genes under different stresses), is missing some input on possible mechanisms. The authors replied that “As the reviewer suggested, we may consider the addition of more information regarding the roles of Clp proteases on plant abiotic and biotic stresses. However, we should address the lack of knowledge of Clp proteases in plants….Therefore, even in the unsatisfying aspects of this manuscript, please accept as it is because new plant scientists would have more interest in this amazing regulatory machinery in chloroplast and further develop studies”.
I can understand that the purpose of the review is then to address the lack of general knowledge of Clp proteases, however the title suggests that the protective functions against abiotic stress and pathogen invasion, as well as the underlying mechanisms, should be the focus of the review.
The identification of substrates of Clp proteases is certainly useful to suggest targeted processes, as the authors also state in lines 361-365. There are a few comprehensive studies that should be integrated:
Boris Zybailov, Giulia Friso, Jitae Kim, Andrea Rudella, Verenice Ramírez Rodríguez, Yukari Asakura, Qi Sun and Klaas J. van Wijk, Large Scale Comparative Proteomics of a Chloroplast Clp Protease Mutant Reveals Folding Stress, Altered Protein Homeostasis, and Feedback Regulation of Metabolism, Molecular & Cellular Proteomics August 1, 2009, First published on May 7, 2009, 8 (8) 1789-1810; https://doi.org/10.1074/mcp.M900104-MCP200
Jui-YunRei Liao, Klaas J.van Wijk, Discovery of AAA+ Protease Substrates through Trapping Approaches, Trendi in Biochemical Sciences, Volume 44, Issue 6, June 2019, Pages 528-545
Other corrections:
Line 64: caseinolytic and not “casenolytic”
